



# Interannual and seasonal variations in aerosol optical depth of the atmosphere in two regions of Spitsbergen Archipelago (2002-2018)

Dmitry M. Kabanov[1], Christoph Ritter[2], Sergey M. Sakerin[1]

[1]V.E. Zuev Institute of Atmospheric Optics, Siberian Branch, Russian Academy of Sciences, Academician Zuev Square 1,
Tomsk, Russia
[2]Alfred Wegener Institute for Polar and Marine Research, 14473 Potsdam, Germany

*Correspondence to*: Christoph Ritter (christoph.ritter@awi.de)

**Abstract.** In this work hourly averaged sun photometer data from the sites Barentsburg and Ny-Ålesund, both located in Spitsbergen in the European Arctic, are compared. Our data set comprises the years 2011 to 2017. We found for more turbid periods (aerosol optical depth $\tau_{0.5} > 0.1$) that typically Barentsburg is more polluted than Ny-Ålesund, especially in the short wave spectrum. However, the diurnal variation of AOD is highly correlated. Next, $\tau$ was divided into a fine and coarse mode. It was found that generally the fine mode aerosol optical depth dominates and also shows a larger interannual as inner annual variation. Tau fine $\tau^f$ is in fact larger in spring during the Arctic Haze period. Overall the aerosol optical depth seems to decrease, although this is not statistically significant.

## 1 Introduction

The studies of the character and causes of variations in all components of climate system, including the aerosol composition of the atmosphere, became more urgent with regard to climate change [IPCC, 2013]. Atmospheric aerosol plays an important role in the processes of solar radiative transfer and exchange by different substances (and, in particular, pollutants) between the continents and ocean [e.g. Kondratyev et al., 2006]. As compared to gases, aerosol is characterized by a complex physicochemical composition and stronger variation of concentration and radiative impact.

Of the various aerosol characteristics, the observations of aerosol optical depth (AOD) of the atmosphere are most widespread and carried out at the international and national networks of stations using sun photometers (see, e.g., [WMO, 2005; Holben et al., 1998]). The AOD represents the extinction of radiation, integrated over atmospheric column, and can be considered as an optical equivalent of the total aerosol content.

One of the main aerosol climatology problems is to determine the specific features of interannual and seasonal variations in different regions. Multiyear observation time series no shorter than 10 years are required to clarify these regularities. More specifically, under the conditions of the changing climate system, even 20 years may turn out to be insufficient to identify the tendencies or certain periodicities in the multiyear variations of aerosol characteristics.



First observations of spectral AOD in the Arctic zone were carried out about 40 years ago [Shaw, 1982; Freund, 1983; Radionov and Marshunova, 1992]; however, they become regular in character only in the early 2000s, after the development of photometric observations at continental stations. A comprehensive overview of atmospheric AOD in the Polar Regions has been presented by C. Tomasi [Tomasi et al., 2012; 2015].

Studies by various authors showed that the Arctic atmosphere is affected appreciably by outflows of aerosols of different types
(smoke, industrial, sulfate, organic) from Eurasia and North America. The most powerful effect is due to smoke from forest fires that cover large areas of boreal zone [e.g. Chubarova et al., 2012; Sitnov et al., 2013; Zhuravleva et al., 2017]. Long-range transports of smoke plumes lead to a considerable pollution of the Arctic atmosphere [Stohl et al., 2006; Stone et al., 2008; Eck et al., 2009; Vinogradova et al., 2015]. These episodes are short-term (1-3 days) and rare because they depend on the product of probabilities of two independent events: (a) the fire itself in any area of boreal zone, and (b) the fact that the
trajectory of air transport from a fire center arrived exactly at a given region of the Arctic.

In addition to smokes, anthropogenic and other types of fine aerosol also outflow from midlatitudes. In contrast to forest fires, the sources of these aerosols operate almost all the time and are distributed over the entire territory of human life activity. A somewhat larger concentration of anthropogenic aerosol in densely populated regions of Europe was observed in the past century; however, recently the industrial emissions have been stabilized or reduced in this area [Tørseth et al., 2012; Li et al.,
2014; Zhdanova et al., 2019].

An AOD increase may also be associated with volcanic activity. In the period of time considered here, there were no large eruptions (like Pinatubo volcano in 1991), having a global effect. The effects of less powerful volcanic eruptions on the Arctic atmosphere are short-term (a few days in duration) and comparable to those due to smokes from forest fires. For instance, AOD increase on Spitsbergen was observed after the eruptions of volcanoes Kasatochi (August 2008) and Sarychev (July
2009) [Hoffmann et al., 2010; Toledano et al., 2012].

The effects of pollutant outflows on the Arctic atmosphere intensify in late winter – early spring. The temperature inversions in this season lead to the formation and accumulation of aerosol in separate layers of the troposphere (the Arctic Haze Phenomenon) [e.g. Shaw, 1995; Quinn et al., 2007; Tomasi et al., 2015].

In the recent decade increasingly more work was published analyzing the multiyear AOD variations in different regions, either
based on spectral [Zhdanova et al., 2019; Chubarova et al., 2016; Putaud et al., 2014; Li et al., 2014; Xia, 2011; Michalsky et al., 2010; Sakerin et al., 2008a; Weller et al., 1998] or integrated (actinometric) [Gorbarenko and Rublev, 2016; Plakhina et al., 2009; Ohvril et al., 2009] measurements of solar radiation. In most regions of Eurasia the AOD shows a decreasing tendency following 1995. The negative AOD trend is associated with decreasing anthropogenic load and the absence of powerful volcanic eruptions analogous to Pinatubo (July 1991) and El Chichón (March – April 1982). Of particular interest is
the dynamics of aerosol constituent of the atmosphere in the Arctic zone where there are the largest climate changes [IPCC, 2013]: increased temperature and prolonged warm period, reduction of sea-ice area, changes in the circulations, etc.

The archipelago of Spitsbergen in the European Arctic is a special region because it is strongly influenced by the warm West Spitsbergen Current. Hence, for the given geographical latitude the West Coast of Spitsbergen faces warm and moist



conditions. Moreover, Spitsbergen currently faces a pronounced winter warming of about 2 degrees per decade, which can partly be explained by more efficient advection from the Atlantic Ocean (Dahlke and Maturilli, 2017). For this reason aerosol properties over Spitsbergen may not be directly comparable to aerosol observations at other Polar sites. On the other hand, we may see aerosol properties at Spitsbergen now which are more typical for the Arctic in the future in warmer, more marine conditions.

For Spitsbergen an analysis of aerosol properties for separate periods was already performed before [Herber et al., 2002; Glantz et al., 2014; Chen et al., 2016; Pakszys and Zielinski, 2017]. In this work, we discuss the AOD measurements in 2002-2018 in two regions of Spitsbergen: Ny-Ålesund and Barentsburg. Based on the multiyear observation time series, we considered the following issues: (a) differences in AOD between neighboring regions; (b) choice of the method for extracting the contributions of two (fine and coarse) AOD components; (c) seasonally average AOD variations during the polar day; and (d) the specific features of the interannual AOD variations during two periods of measurements (2002-2010 and 2011-2018).

## 2 Instruments, methods, and observational data

### 2.1 Characterization of instruments and regions of measurements

Historically, observations of the spectral AOD $\tau^a(\lambda)$ on Spitsbergen Archipelago are carried out at three closely lying scientific stations (in the order of decreasing latitude): Ny-Ålesund (78°54′N, 11°53′E), Barentsburg (78°04′N, 14°13′E), and Hornsund (77°00′N, 15°34′E). The distances from Barentsburg to Ny-Ålesund and Hornsund are 110 and 120 km respectively.

Measurements of atmospheric AOD in the scientific settlement Ny-Ålesund (population of about 100 residents during summer) have been performed since 1991. At the first stage (1991-1999) the observations were performed in separate periods of the year (polar day, polar night) using photometers of different types (Sun, Moon, Star). The results of those studies were considered in [Herber et al., 2002]. Here, we analyze the AOD variations in a later period when measurements became regular and homogeneous in character. The main characteristics of sun photometers (SP1A and SP2H), used in measurements, are presented in Table 1.

The sun photometer in Ny-Ålesund is located just south of the settlement in about 10m asl on the BSRN radiation field. The temporal resolution of the data is 1 minute. The instruments are regularly calibrated in Izaña /Tenerife. The air masses for aerosol and for ozone were considered according to the formulas by Kasten and Young (1989) and the WMO No 183 report (2008), respectively. A cloud screening similar to Alexandrov et al. (2004) has been performed.



Table 1. Characteristics of Sun-photometers and data volume in Ny-Ålesund and Barentsburg

| Scientific stations | Sun-photometers | Angle of view, degrees | Range of spectrum // number of spectral channels | Number of hours (days) of measurements |
|---|---|---|---|---|
| Ny-Ålesund (AWI, Germany) | SP1A, SP2H | 1 | 0.37-1.06 μm // 13 <br><br> 0.37-1.05 μm // 12 | 7520 (1130) |
| Barentsburg (AARI and IOA SB RAS, Russia) | SPM, SP-9 | < 2,5 <br><br> < 2 | 0.34-2.14 μm // 11 <br><br> 0.34-2.14 μm // 13 | 1732 (350) |

In 2011 the monitoring of aerosol characteristics (including AOD [Sakein et al., 2018a]) was organized in the Russian Scientific Center, located in the southwestern part of Barentsburg settlement on the coast of Grønfjorden Gulf. Products of coal-mining industry and thermal power plant, located at distance of about 1 km, may influence the aerosol characteristics in

Barentsburg (population of about 500 residents).

The atmospheric AOD was initially measured using SPM portable sun photometer [Sakein et al., 2013], altitude of 65 m above sea level. In 2015 it was changed to SP-9 photometer with automatic Sun tracking system (the instrument was installed at the altitude of 74 m). The base set of the wavelength channels comprises the interference filters with the passband maxima at the wavelengths: 0.34, 0.37, 0.41, 0.50, 0.55, 0.67, 0.77, 0.87, 1.04, 1.25, 1.55, and 2.14 μm. Still another wavelength channel

(0.94 μm) is used to determine the total water vapor content of the atmosphere.

The method for calculating the spectral AOD [Kabanov and Sakerin, 1997; Kabanov et al., 2009] includes accounting for the spectral transmission functions of light filters, and Rayleigh scattering and absorption by atmospheric gases: $H_2O$, $O_3$, $CO_2$ and others. The absorption is calculated using the databases of spectral line parameters HITRAN-2000 (http://www.hitran.com), models of continual absorption MT_CKD (http://rtweb.aer.com/continuum_code.html) and vertical

gas distribution AFGL [Anderson et al., 1986]. Water vapor absorption is accounted for using real water vapor contents, measured in the wavelength channel of 0.94 μm.

Total amount of data (hours and days of measurements), which were used in the AOD analysis in two regions, is presented in Table 1. Seasonal and interannual AOD variations were analyzed for the period of polar day (March-September). First the hourly average AOD values were used to calculate the averages for each day of measurements, then the monthly averages

were calculated, based on which the averages for each year (or, more specifically, from March to September) were determined. For brevity, they will be called Daily, Monthly, and Annual AOD. The interannual variations in AOD were estimated in two variants: (a) according to the average values in the measurement period (Annual AOD); and (b) according to the averages in


periods of spring maximum and autumn minimum of AOD.


## 2.2 Data comparison and preparation of observation time series

A comparison of observations using two photometers may be of interest for intercalibration of the instruments, i.e., estimating instrumental-methodic AOD determination errors. If the measurements are separated in space (as in the given case), the difference in the data makes it possible to estimate the local AOD inhomogeneities, caused by anthropogenic or natural factors: local weather conditions, type and state of the underlying surface, orography, and the effect of industrial or other sources of aerosol.

It should be noted that we already compared earlier the AOD measurements at the neighboring stations on Spitsbergen Archipelago, i.e., Hornsund and Ny-Ålesund [Toledano et al., 2012; Pakszys and Zielinski, 2017]. Comparison of time independent measurements showed that the average difference in the annual and seasonal AOD values at the wavelength of 0.55 μm does not exceed 0.01-0.02. Larger AOD values occur in periods of Arctic haze and outflows of smokes from forest fires, which are differently manifested in these two regions.

In contrast to the above-mentioned works, we compared quasi-synchronous (nearly time coincident) AOD measurements in the neighbouring regions in 2011-2017 (see [Kabanov et al., 2018] for more details). The data of the SP1A (Ny-Ålesund) and SPM (Barentsburg) photometer observations were used to calculate the hourly average AOD values. Then, the datasets from the two regions were "juxtaposed" provided that the times of the AOD measurements differed by no more than one hour.

Comparison of measurements with the two photometers showed a large dispersion of the data under the conditions of strong atmospheric turbidities, i.e., during forest fire smoke outflows from Siberia or Alaska. In these rare situations, the aerosol characteristics stand out in larger spatial inhomogeneity, so that the comparison becomes incorrect. Therefore, further analysis was performed for usual situations, when $\tau^a$ (0.5 μm) < 0.2. Figure 1 illustrates the regression relation between $\tau^a$ (0.5 μm) measurements in the neighbouring regions of Spitsbergen.

A comparison of the statistical characteristics showed that the average AOD values are a little larger in Barentsburg than in Ny-Ålesund. The maximal difference in AOD is observed in the shortwave part of the spectrum (0.38 μm), $\Delta = \tau^a$ (SPM) – $\tau^a$ (SP1A) = 0.024; while in IR range (0.87μm) the difference decreases to $\Delta = 0.005$. This feature indicates that fine aerosol is more abundant in the atmosphere of Barentsburg. At the same time, we note that the AOD differences are minor (comparable with the error), and the interdiurnal AOD variations in the two regions are coordinated in character (correlation coefficients are 0.83-0.89). Comparison of quasi-synchronous AOD measurements in Barentsburg and Hornsund gave close results [Kabanov et al., 2018]: $\Delta = 0.004 – 0.024$, the correlation coefficients are 0.71-0.81. Hence, observations in the neighbouring regions on Spitsbergen are quite compatible and identically reflect the specific features of the AOD variations.


The joint use of results from AOD monitoring in the neighbouring regions makes it possible to control the reliability of information, as well as to identify the specific features of AOD variations not only for a specific site, but for the region as a whole. The results of the observations in each of the regions have their own advantages. The advantage of the data from Barentsburg (SPM / SP-9 photometers) is a wider range (0.34-2.14 µm) of the spectral measurements and the possibility to separate the contributions from two AOD components, using an empirical method (see subsection 2.3).

The valuable feature of the data from Ny-Ålesund is a longer AOD observation time series. However, different errors have been accumulated in these data for the long period of measurements. A simple exclusion of all suspect AOD measurements was undesirable because for analysis of multiyear variations it was necessary to keep the observation time series as long as possible. Taking this circumstance into account, the multiyear observation time series was prepared for sorting out or correction of suspect AOD values [Kabanov et al., 2019a]. The initial dataset was processed to remove the data in which short-term bursts

or rapid AOD variations were observed, as well as the distortions to smoothness of the wavelength dependences $\tau^a(\lambda)$. Owing to a certain redundancy of the number of spectral channels, we could identify false measurements and select most reliable data.

### 2.3 Fine and coarse AOD components

In analysis of variations in the spectral AOD of the atmosphere, the parameters α and β of the Ångström formula are widely used:

$$\tau^a(\lambda) = \beta \cdot \lambda^{-\alpha},$$
(1)

where β is the turbidity coefficient, which is close in value to AOD at the wavelength of 1 µm; and α is the selectivity exponent, characterizing the spectral decrease in AOD.

Numerous studies in different regions and atmospheric conditions showed that the formula (1) does describe well the wavelength dependence $\tau^a(\lambda)$ in the main range (0.34 – 1 µm) of AOD measurements. At the same time, the use of this formula

has limitations and disadvantages, requiring an explanation.

First, the Ångström formula becomes unsuitable for describing the wavelength behavior of AOD in the atmospheric "transparency windows" in the wavelength range of 1 – 4 µm. This is because the power-law dependence (1) stems from the combined action of fine and coarse aerosol fractions, which have different spectral properties. Extinction of radiation by small particles ($2\pi r/\lambda < 1$) is dominant in the visible region of spectrum; however, it rapidly decays with the growing wavelength

and becomes insignificant in the region beyond of 1 µm. Extinction of radiation by coarse aerosol barely changes with the wavelength and becomes predominant in the IR range. Mie calculations and experimental data [Sakerin and Kabanov, 2007a; Sakerin et al., 2008b] confirm that the power-law AOD decay gradually goes over into almost neutral dependence. Therefore, $\tau^a(\lambda)$ in a wider wavelength range is better to represent by a sum of two components:

$$\tau^a(\lambda) = \tau^c + \tau^f(\lambda) = \tau^c + m \cdot \lambda^{-n},$$
(2)





where $\tau^c$ is the constant (wavelength independent) coarse AOD component; $\tau^f(\lambda)$ is the selective fine component; and *m* and *n* are the parameters analogous to those in the Ångström formula.

Second, the Ångström parameters do not allow one to interpret the causes for AOD variations unambiguously. An increase/decrease in the exponent $\alpha$ is sometimes unjustifiably attributed only to changes in fine aerosol. In fact, the exponent $\alpha$ conceals the actions of a few factors. The wavelength dependence of AOD is indeed determined by the optical properties of

fine component $\tau^f(\lambda)$. Both the sizes and refractive index of small particles influence the degree of wavelength decay of AOD (and values of the parameters *n* and $\alpha$). But, precisely what caused changes in the selectivity of AOD is almost impossible to determine without the use of independent data.

The next uncertainty factor is that the exponent $\alpha$ depends on the *relationship* of the optical depths of fine and coarse aerosol ($\tau_\lambda^f / \tau^c$). That is, $\alpha$ may increase both due to growing content of fine aerosol, and to decreasing content of coarse aerosol.

The presence of the interrelation between $\alpha$ and ($\tau_\lambda^f / \tau^c$) is illustrated in Fig. 2. The correlation coefficient between $\alpha$ and ln

$[(\tau_\lambda^f / \tau^c) + 1]$ is statistically significant and equal to 0.68 (P < 0.0001).

We also note that the component $\tau^c$, which influences the exponent $\alpha$, is tightly related and has close values to the second Ångström parameter [Sakerin and Kabanov, 2007a, b]: $\tau^c \approx \beta$. A consequence of this is that the parameters $\alpha$ and $\beta$ are themselves correlated.

Thus, the use of the Ångström parameters in the analysis of AOD variations is ambiguous and may lead to erroneous conclusions. It is more preferable to consider the specific features of variations in two AOD components: $\tau^f(\lambda)$ and $\tau^c$. In addition to different sizes and spectral properties, fine and coarse aerosol fractions differ in the origins of particles and their transformation in the atmosphere. Fine (sulfate, organic, etc.) aerosol is formed in the atmosphere as a result of various photochemical and microphysical processes [Kondratyev et al., 2006]. The lifetime of fine aerosol is a few days; therefore, it

can be transported long distances (hundreds and thousands of kilometers) away. The main source of coarse (marine and dust) aerosol is the underlying surface. Because of its short lifetime and small transport distance, coarse aerosol is more local in character and pertains to a specific terrain. The only exceptions are powerful dust outflows in tropical latitudes.

**2.4 Methods for determining fine and coarse AOD components**

As was already indicated above, in the IR range, the effect of fine aerosol becomes insignificant, and AOD is determined only by the coarse component. Therefore, $\tau^c$ can be determined by *empirical* method (EM), i.e., from minimal or average AOD values, measured in the range of 1.24-2.14 μm [Sakerin and Kabanov, 2007b; Sakerin et al., 2008b]. Then, the second (fine) component is found as a residual of the total AOD. Usually, it is estimated for the wavelength of 0.5 μm: $\tau_{0.5}^f = \tau_{0.5}^a - \tau^c$.



However, most sun photometers (and in particular the SP1A in Ny-Ålesund) operate in the wavelength range up to 1.05 μm,

making *empirical* method inapplicable. In this case, $\tau^c$ and $\tau^f_{0.5}$ can be estimated using calculation methods. For instance, in

the AERONET system (http://aeronet.gsfc.nasa.gov), $\tau^f_{0.5}$ is calculated using *spectral deconvolution algorithm* [O'Neill et al.,

2003], based on the relationship of spectral AOD, measured in the shortwave part of the spectrum 0.38-1.02 μm.

In the work [Kabanov and Sakerin, 2016] we suggested simpler methods for separating the contributions from $\tau^f_{0.5}$ and $\tau^c$,

based on the regression interrelations with the parameters of Ångström formula. Comparison of different methods of $\tau^f_{0.5}$ ($\tau^c$)

estimation for the conditions of the marine and continental (Tomsk) atmosphere showed close results: the average difference

of $\tau^c$ from data of base *empirical* method does not exceed 0.007 for the standard deviation from 0.006 to 0.024.

For the conditions of Arctic region (Spitsbergen), we performed an additional study [Kabanov et al., 2019], concerning the

selection of an optimal method of $\tau^f_{0.5}$ ($\tau^c$) estimation. Different methods were tested using SPM photometer measurements

of AOD in Barentsburg. The error of the methods was estimated by comparing the calculated values of $\tau^f_{0.5}$ or $\tau^c$ with the data

from base *empirical* method (EM).

Figure 3 illustrates the results of testing two regression methods (RM1 and RM2), based on the interrelations (a) between $\tau^c$

and the parameter $\beta$, and (b) between $\tau^f_{0.5}$ and the parameters $\alpha$, $\beta$. For the conditions of Spitsbergen, we obtained the following

regression equations:

RM1: $\tau^c = 0.665 \cdot \beta - 0.0005$           (3)

RM2: $\tau^f_{0.5} = (-0.829 + 1.066 \cdot 0.5^{-\alpha}) \cdot \beta$        (4)

Table 2 presents the standard deviations σ and the correlation coefficients R between the calculated (RM1, RM2) and empirical

(EM) $\tau^c$ values. These results suggest the regression methods make it possible to estimate $\tau^c$ with an identical error of 0.007.

Table 2. Estimates of applicability of different methods (RM1, RM2, IM1, IM2) of the $\tau^c$ ($\tau^f_{0.5}$) calculation

| Parameter | RM1 | RM2 | IM1 | IM2 |
|---|---|---|---|---|
| σ | 0.007 | 0.007 | 0.007 | 0.008 |
| R | 0.819 | 0.963 | 0.967 | 0.953 |

The disadvantage of the regression methods is that they require a preliminary data accumulation under the conditions of a

specific region for determining optimal regression coefficients in equations (3) and (4). Of course, the error of the regression

methods may increase if aerosol characteristics strongly differ from those typical for the region and do not correspond to the

selected regression coefficients.

Therefore, in addition to the regression methods, we considered the applicability of another two methods of $\tau^f_{0.5}$ estimation,

based on the results of solving the inverse problem, namely: retrieval of particle sizes from measurements of spectral AOD.





The inversion method 1 (IM1) is based on the interrelation between $\tau_{0.5}^{f}$ and volume or cross section of particles of fine aerosol. This method is implemented in the following steps: (a) based on spectral AOD, an inversion algorithm is used to retrieve the particle distribution functions ($dS/dr$) or ($dV/dr$); (b) the radius range of particles of fine fraction is chosen, for which the total area ($S^{f}$) or volume ($V^{f}$) of particles is calculated; and (c) based on regression relations, parameters of approximation of the dependence of $\tau_{0.5}^{f}$ (EM) on the calculated $S^{f}$ or $V^{f}$ are selected.

The inversion method 2 (IM2) is implemented by solving first the inverse problem, and then the direct problem of the aerosol optics: (a) as in IM1, the spectral AOD values are used to retrieve the distribution functions ($dS/dr$); and (b) based on the ($dS/dr$) values, $\tau_{0.5}^{f}$ is calculated for the size range of fine aerosol.

The inverse problem on retrieving the distribution functions ($dS/dr$) was solved using iteration algorithm of M.A. Sviridenkov [Sviridenkov, 2001], modified from Twitty algorithm [Twitty, 1975]. The particle distribution was assumed to be lognormal, and the refractive index was assumed to have the real part of 1.5 and the imaginary part of 0. Applicability of inversion methods was estimated for a few variants: (a) for different wavelength intervals of AOD (0.34-2.14 μm; 0.38-0.87 μm; 0.38-1.02 μm; and 0.34-1.02 μm); (b) for the distribution functions ($dS/dr$) and ($dV/dr$); and (c) for different radius boundaries of particles of fine fraction (0.1-0.5 μm and 0.1-0.45 μm). Figure 4 presents examples of interrelations: (a) between $\tau_{0.5}^{f}$ (EM) and calculated values of particle volume $V^{f}$; and (b) between $\tau_{0.5}^{f}$ values, determined using base (EM) and inversion (IM1) methods. The calculations in this case were performed for the wavelength range of AOD 0.38-1.02 μm and particle radius range of 0.1-0.5 μm.

Analysis of application of IM1 and IM2 [Kabanov et al., 2019a] showed that the $\tau_{0.5}^{f}$ determination error decreases by about a factor of 1.5 when the AOD is used in a wide (0.34-2.14 μm) wavelength range. However, for the narrower wavelength range of the SP1A photometer (0.38-1.02 μm) the $\tau_{0.5}^{f}$ calculation error is comparable with results from regression methods (see Table 2). That is, the relative errors of the $\tau^{c}$ and $\tau_{0.5}^{f}$ calculations for the mean conditions of Barentsbutg ($\tau_{0.5}^{a} = 0{,}086$ [Sakerin et al., 2018a]) are 30% and 11% respectively.

The IM1 method was chosen for a subsequent use. Despite a more complicated procedure of its calculations, IM1 is more sensitive to aerosol variations, which is indicated by the highest correlation coefficient between $\tau_{0.5}^{f}$ (IM1) and data from base method (EM).

There may be a question as to why after retrieval of aerosol disperse composition we nonetheless consider the seasonal and interannual variations in optical characteristics: $\tau^{c}$ and $\tau_{0.5}^{f}$? Analysis of disperse composition of aerosol is a more complex and non-unique problem because it is necessary to consider the transformation of two aerosol fractions, which are described by a few parameters: shapes and widths of distributions for each fraction, separation boundary, and effective particle radii. Moreover, an uncertainty remains about the values of these parameters because of the priori specified aerosol refractive index.



In this work, we pursued a simpler task: to determine the character and magnitude of variations in aerosol optical characteristics. In this case, instead of many microstructure parameters, it is sufficient to analyze their more compact optical image in the form of two components, $\tau^f$ and $\tau^c$.

### 3. Discussion of the results

Current climate change and environmental transformation influence the regularities of variations in aerosol characteristics to some degree. Because of the deficit of its own aerosol sources in the Arctic zone, an important role in AOD variations is played by outflows of smoke, anthropogenic and volcanic aerosol from midlatitudes. The frequency of these outflows in particular months and years determines the specific features of seasonal dynamics of AOD in Arctic regions and magnitude of interannual oscillations.

### 3.1 Interannual variations

The highest atmospheric turbidities in the region of Spitsbergen were observed on July 10, 2015 and on May 2-3, 2006 (Fig. 5). Daily AOD (0.5 μm) in these cases reached 0.82 and 0.6, about an order of magnitude lager than multiyear averages. After passing to monthly AOD values, the effect of these short-term turbidities decreased to 0.192 in May 2006 and 0.152 in July 2015. Trajectory analyses of air mass motion showed that the extreme AOD values in July 2015 were due to long-range transport of smokes from forest fires in Alaska [Sakerin et al., 2018a, Pakszys and Zielinski, 2017; Markowicz et al., 2016]. We also considered in detail the second anomalous situation (in May 2006) [e.g. Myhre et al., 2007; Stohl et al., 2007], associated with the outflow of smoke aerosol from agricultural fires in the Eastern Europe.

Episodes with high atmospheric turbidities were also observed in June 2003, March and August 2008, in April and August 2009, and in April 2011. The AOD values in these periods of time had already been analyzed by many authors [Toledano et al., 2012; Glantz et al., 2014; Tomasi et al., 2015; Chen et al., 2016; Pakszys and Zielinski, 2017]. Independent of the causes for these short-term turbidities (Arctic haze, outflows of smoke or volcanic aerosol), they enhance not only the monthly, but also annual AOD values.

The above-mentioned high-turbidity episodes (2006, 2008, 2009, 2015) were reflected partly in annual AOD oscillations (Fig. 6). Moreover, a maximum appeared in the interannual variations in 2011-2012. This maximum was due not to extreme 1-3-day AOD bursts, but to stronger turbidities as compared to the neighbouring years.

The annual AOD maxima occur with the average periodicity of about three years. When high-turbidity episodes are eliminated (see dashed line in Fig. 6), certain maxima disappear; however, the general character of the AOD oscillations remains unchanged. Among these maxima, the highest AOD value in 2003 seems suspect. This annual AOD value cannot be considered as representative because of short period of observations (4 days in March and 5 days in May-June) in this year.

In addition to oscillations, a tendency for a minor AOD decrease can be discerned in the multiyear variations. This is also indicated by a comparison of AOD characteristics in two periods of observations: the average AOD (0.5 μm) in 2011-2018





decreased by 0.013 relative to 2002-2010 (Fig. 5). However, this decreasing AOD tendency is not statistically significant. The statistical estimates of $\tau_{0.5}^f$ and $\tau^c$ variations in spring and fall periods in these two regions (Fig. 7) also revealed no trend

component.

From the statistical characteristics (Table 3) and Fig. 7 it follows that AOD and its interannual variations are determined mainly by fine aerosol: the SD values and the range <Min – Max> are a factor of 2-3 larger for $\tau_{0.5}^f$ than $\tau^c$. The relative $\tau_{0.5}^f$ and $\tau^c$ variations (see the variation coefficients V) are 14-29%. No explicit predominance of the variation coefficients for any AOD component can be seen. For instance, in the period of 2011-2018, the variation coefficients were larger for $\tau_{0.5}^f$ than $\tau^c$ (25

and 20%) in Ny-Ålesund and with a reverse relationship (14 and 23%) found in Barentsburg.

The interannual oscillations in the Ångström exponent can be considered as minor: the variation coefficients for $\alpha$ are 13-15%. The average $\alpha$ and the total variability range (from 1 to 1.7) are in the regions of values characteristic for the continental midlatitude atmosphere and are larger than in the marine atmosphere [Sakerin et al., 2008b; 2018b]. These values of the Ångström exponent are because the ratio ($\tau_{0.5}^f / \tau^c$ = 2.6-2.9) and the relative contribution of fine component ($\tau_{0.5}^f / \tau_{0.5}^a$ = 0.73-

0.75) are close to continental values. As an example, we present multiyear data in boreal zone of Siberia in spring (smoke-free) period [Kabanov et al., 2019b]. The average AOD values in Siberia are about two times larger ($\tau_{0.5}^f$ = 0.105, $\tau^c$ = 0.036) than in Ny-Ålesund, and the ratio ($\tau_{0.5}^f / \tau^c$) and the exponent $\alpha$ are almost the same: ($\tau_{0.5}^f / \tau^c$) = 2.92 and $\alpha$ = 1.43.






Table 3. Statistical characteristics of Annual AOD: average, minimal (Min), maximal (Max) values, standard deviations (SD),
and variation coefficients (V); values for Ny-Ålesund (2002-2018) in the first row, for Ny-Ålesund (2011-2018) in the second
row; and for Barentsburg (2011-2018) in the third row

| Characteristics | Mean | SD | Min | Max | V, % |
|---|---|---|---|---|---|
| | | | | | |
| $\tau_{0.5}^{a}$ | 0.067 | 0.017 | 0.04 | 0.10 | 25 |
| | 0.059 | 0.012 | 0.04 | 0.08 | 20 |
| | 0.080 | 0.007 | 0.07 | 0.09 | 10 |
| $\tau_{0.5}^{f}$ | 0.050 | 0.013 | 0.02 | 0.07 | 26 |
| | 0.044 | 0.011 | 0.02 | 0.06 | 25 |
| | 0.058 | 0.008 | 0.04 | 0.07 | 14 |
| $\tau^{c}$ | 0.017 | 0.005 | 0.01 | 0.03 | 29 |
| | 0.015 | 0.003 | 0.01 | 0.02 | 20 |
| | 0.022 | 0.005 | 0.01 | 0.03 | 23 |
| $\alpha$ | 1.41 | 0.19 | 1.03 | 1.71 | 13 |
| | 1.44 | 0.21 | 1.11 | 1.71 | 15 |
| | 1.24 | 0.19 | 1.01 | 1.60 | 15 |
| $\beta$ | 0.026 | 0.008 | 0.016 | 0.040 | 31 |
| | 0.022 | 0.006 | 0.016 | 0.034 | 27 |
| | 0.035 | 0.005 | 0.025 | 0.042 | 14 |


From Figs. 6 and 7 it is clearly seen that AOD in Barentsburg is almost always higher than in Ny-Ålesund. The average excess
of annual AOD is (see rows 2 and 3 in Table 3): $\Delta\tau_{0.5}^{a} = 0.02$, $\Delta\tau_{0.5}^{f} = 0.012$, $\Delta\tau^{c} = 0.008$. This result indicates that a little
more anthropogenic (fine) and coarse aerosols are contained in the atmosphere of Barentsburg (see also Figs. 1 and 5).

Variations in annual AOD in Ny-Ålesund and Barentsburg are coordinated in character. Oscillations in fine component
sometimes show opposite changes in the two regions, such as in spring 2013 (see Fig. 7). Different behaviors of $\tau_{0.5}^{f}$ may be
because observation time series are inhomogeneous in each region due to clouds or because AOD are measured at different
times.



### 3.2 Specific features of seasonal variations

The most common regularity of the seasonal AOD behavior at midlatitudes is the spring (and sometimes also summer) maximum and fall minimum [e.g., Sakerin et al., 2015; Chubarova et al., 2014; Holben et al., 2001]. The primary causes for this AOD behavior are the annual cycle in the Sun's declination meaning a return of sunlight and possibly a longer aerosol life-time over the frozen ocean. The springtime increases in insolation and temperature trigger a few processes: (a) snow cover evaporates and melts; (b) the atmosphere is enriched by different deposition products, accumulated over the winter, (c) primary (marine, soil) aerosol starts to come from the underlying surface; and (d) photochemical processes of production of in situ

aerosol in the atmosphere and emission of organic aerosol are activated [e.g., Kondratyev et al., 2006].

The seasonal AOD dynamics in the Arctic zone is analogous to midlatitudes: springtime maximum and a decay toward fall [e.g., Toledano et al., 2012; Tomasi et al., 2015; Sakerin et al., 2018]. This AOD behavior is because of similar annual rhythms of both own aerosol sources in the Arctic and long-range aerosol transports from midlatitudes.

Figure 8 shows the seasonal variations in monthly AOD for two regions in Spitsbergen. The seasonal AOD behaviours in Ny-

Ålesund were similar in character in 2002-2010 and 2011-2018. The difference is that AOD values in March-June have decreased by ~0.02 in recent 8 years. At the same time, monthly AOD in the second half of polar day (July-September) remained at the same level of low values 0.05-0.06. As a result, the seasonal AOD decrease in 2011-2018 became less pronounced: the relative amplitude had been 30% (versus 55% in the period of 2002-2010).

Seasonal AOD variations in Barentsburg are characterized by an additional summer maximum in July-August. Despite this

difference, common factors in AOD variations in the two regions are nonetheless predominant. Analysis of interrelation between AOD values, measured in Ny-Ålesund and Barentsburg (Fig. 9), showed quite a high (0.90) correlation coefficient. Hence, the synoptic, seasonal, and interannual AOD oscillations are largely coordinated in character.

Observation time series in the two regions were compared to clarify the causes for the summertime AOD maximum in Barentsburg. The increased AOD values in July and August were found to be due to the situations with smoke outflows (and,

in particular, on July 10, 2015 [Sakerin et al., 2018a]). Of the total number of measurements, percentage of smoke-contaminated measurements turned out to be larger in Barentsburg than in Ny-Ålesund. A few rare, but powerful outflows of smoke aerosol have led to an increase in the Monthly $\tau_{0.5}^{f}$ (Fig. 10) and $\tau_{0.5}^{a}$ (Fig. 8) values and distorted the natural seasonal variations. After the events of smoke outflows are eliminated (see dashed lines in Figs. 8 and 10), the seasonal AOD behavior in Barentsburg becomes similar to that in Ny-Ålesund, but with larger (by 0.017, on the average) AOD values. The average

AOD characteristics for the periods of spring maximum and fall minimum in the two regions are presented in Table 4 and in Fig. 11.

Table 4. Average AOD characteristics (± SD) in Ny-Ålesund and Barentsburg during spring (April-May) maximum (first row) and fall (August-September) minimum (second row)





| Characteristics | Ny-Alesund, 2002-2018 | Ny-Alesund, 2011-2018 | Barentsburg, 2011-2018 (No smoke) |
|---|---|---|---|
| $\tau_{0.5}^{a}$ | 0.081±0.03 | 0.070±0.015 | 0.086±0.012 |
| | 0.052±0.023 | 0.052±0.015 | 0.070±0.022 |
| $\tau_{0.5}^{f}$ | 0.062±0.024 | 0.054±0.012 | 0.063±0.012 |
| | 0.038±0.019 | 0.038±0.013 | 0.048±0.02 |
| $\tau^{c}$ | 0.019±0.007 | 0.016±0.005 | 0.022±0.009 |
| | 0.014±0.004 | 0.014±0.003 | 0.022±0.011 |
| $\alpha$ | 1.43±0.23 | 1.47±0.22 | 1.33±0.28 |
| | 1.38±0.26 | 1.35±0.30 | 1.17±0.36 |
| $\beta$ | 0.031±0.01 | 0.026±0.007 | 0.035±0.008 |
| | 0.02±0.008 | 0.02±0.005 | 0.032±0.012 |

From Fig. 10 and Table 4 it can be seen that the fine component makes the major contribution to the formation of the seasonal AOD behavior: the average $\tau_{0.5}^{f}$ values decrease from spring toward fall in the two regions by almost the same amount of 0.015-0.016. Also, modal (most probable) $\tau_{0.5}^{f}$ values vary in a similar way (Fig. 12a). The $\tau_{0.5}^{f}$ mode from spring toward fall shifts from 0.07 to 0.03 in Barentsburg and from 0.05 to 0.03 in Ny-Ålesund. Average (Fig. 10) and modal (Fig. 12b) values of coarse AOD component remain almost unchanged during the polar day: Monthly $\tau^{c}$ values are about 0.015 in Ny-

Ålesund and 0.022 in Barentsburg.

Seasonal decrease of $\tau_{0.5}^{f}$ from spring toward fall leads to changes in the ratio ( $\tau_{0.5}^{f}/\tau^{c}$ ) and spectral AOD dependence (Fig. 11): the slope of the spectral AOD dependence and the Ångström exponent become a little smaller. The relative contribution of fine aerosol to AOD ( $\tau_{0.5}^{f}/\tau_{0.5}^{a}$ ) in Ny-Ålesund is 0.77 in spring and 0.73 in fall. In Barentsburg this ratio is a little smaller, 0.73 and 0.69 respectively.


**4 Conclusions**

We present brief results of our study.

1. It is noted that, to identify the specific features of seasonal and multiyear variations in atmospheric AOD, it is important

to analyze separately fine and coarse AOD components, having different spectral properties, origins, and lifetimes. As applied





to AOD measurements in Ny-Ålesund, we considered a few methods for estimating the contributions of fine and coarse components, and one of the methods (IM1) is selected for a subsequent use. A comparison with data from the base method (EM) showed that the standard deviation of the $\tau^c$ and $\tau^f_{0.5}$ calculations is 0.007, and the relative errors are 30% and 11% respectively.

2. Outflows of fine aerosol of different types from the Eurasian and North American midlatitudes affect appreciably the monthly (and even annual) AOD in the Arctic atmosphere. Outflows of smokes from massive forest and agricultural fires have the strongest effect. The frequency of these episodic outflows in specific years, as well as the frequency of situations of Arctic haze, influence the specific features of seasonal variations and determine the amplitude of interannual AOD variations in the Arctic atmosphere.

3. The oscillations in annual AOD in Ny-Ålesund and Barentsburg are coordinated in character and determined by fine aerosol (interannual variations in $\tau^c$ are a factor of 2-3 smaller). In the multiyear (2002-2018) variations we revealed a tendency of decreasing AOD, but the trend is statistically insignificant. Annual AOD in Barentsburg is, on the average, 0.02 larger than in Ny-Ålesund, indicating larger contents of both fine ($\Delta\tau^f_{0.5} = 0.012$) and coarse ($\Delta\tau^c = 0.008$) aerosol in the bigger settlement.

4. The seasonal AOD variations in Ny-Ålesund are characterized by a decrease from spring toward fall. In the last period
(2011-2018) the seasonal AOD decrease became less pronounced. Monthly AOD values decreased from 0.07-0.09 by the amount of ~0.02 in March-June, and remained unchanged (0.05-0.06) in the second half of polar day. In the seasonal AOD variations in Barentsburg there had been an additional summer maximum, caused by a relatively larger influence of smoke outflows. After smoke outflow events are eliminated, the seasonal behaviour becomes analogous to that in Ny-Ålesund, but with (0.017) higher AOD values.

5. Fine component has the main effect on the formation of seasonal behaviour of AOD. Its relative contribution to AOD ($\tau^f_{0.5}$ / $\tau^a_{0.5}$) is 0.77 in spring and 0.73 in fall in Ny-Ålesund (0.73 and 0.69 in Barentsburg). Coarse AOD component during the polar day is almost unchanged, being, on the average, 0.015 in Ny-Ålesund and 0.022 in Barentsburg. The average annual and monthly values of the Ångström exponent $\alpha$ (from 1.2 to 1.5) do not differ from those at the midlatitudes of the continental atmosphere. That large exponents $\alpha$ are because the ratios of the two AOD components ($\tau^f_{0.5} / \tau^c$) differ little between the Arctic
and continental atmosphere.

**Acknowledgements**

The authors thank all our colleagues who organized and carried out the photometric observations in Ny-Ålesund and Barentsburg, namely, to S. Debatin, J. Graeser, D.G. Chernov, A.V. Gubin, K.E. Lubo-Lesnichenko, A.N. Prakhov, V.F.
Radionov, O.R. Sidorova, and Yu.S. Turchinovich.





The study was supported by the Ministry of Science and Higher Education of the Russian Federation (Project No. AAAA-Ф18-118012500294-9 and AAAA-A17-117021310142-5).

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

**The *Figures***

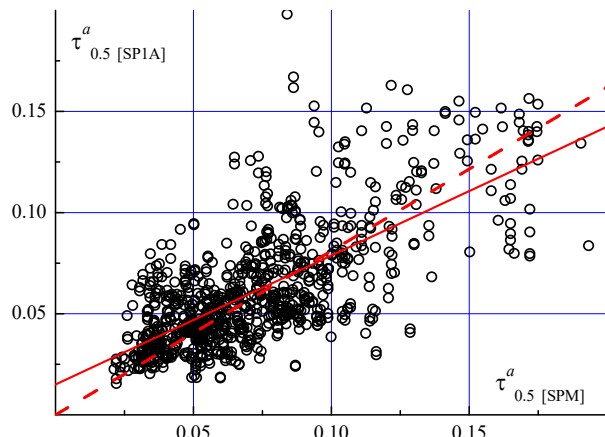

Fig. 1. Scatter diagram of AOD measurements in two regions using SP1A (Ny-Ålesund) and SPM (Barentsburg) photometers

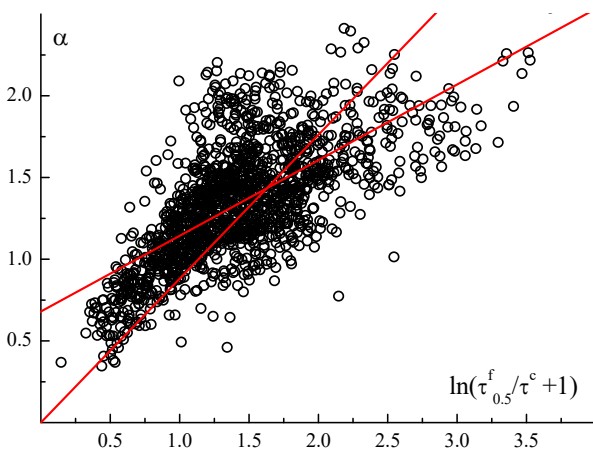

Fig. 2. Correlation dependence between the exponent $\alpha$ and the ratio $\left( \tau_{0.5}^{f}/\tau^{c} \right)$ according to measurements in Barentsburg

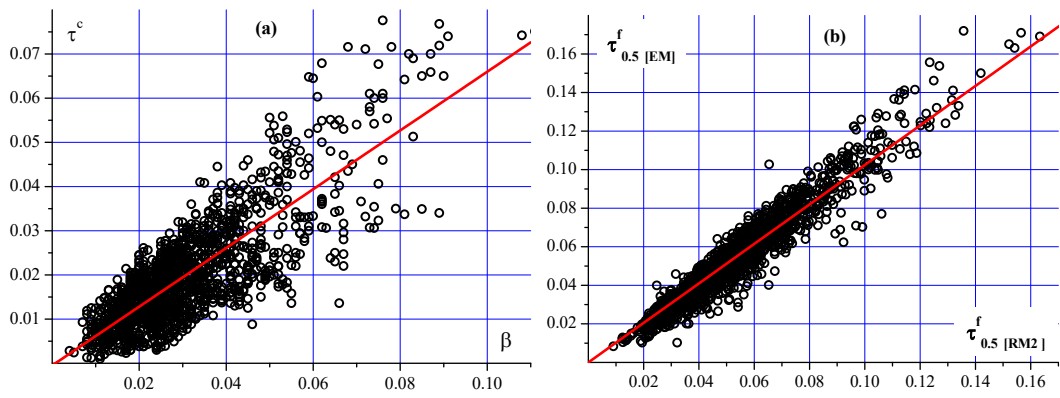

Fig. 3. (a) Interrelation of $\tau^{c}$ with the parameter $\beta$ and (b) interrelation of $\tau_{0.5}^{f}$, calculated using the base (EA) and RM2 methods





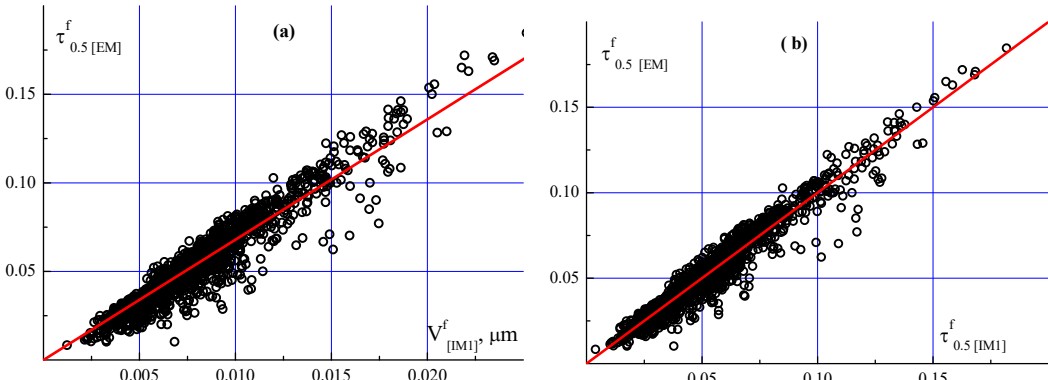

Fig. 4. (a) Interrelation between $\tau_{0.5}^f$ and particle volume $V^f$ and (b) interrelation between $\tau_{0.5}^f$ values, calculated using inversion (IM1) and empirical (EM) methods

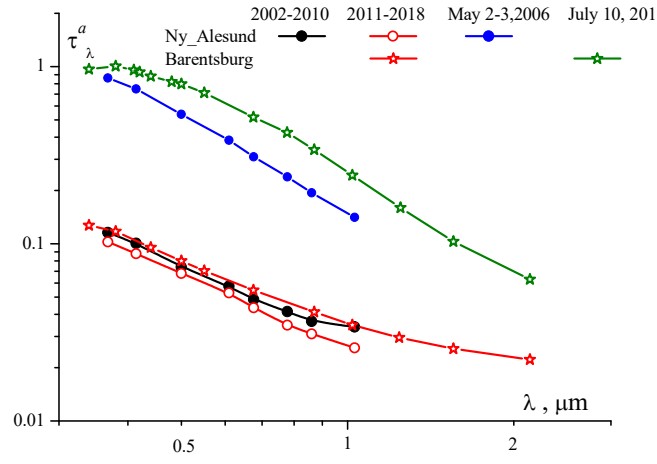


Fig. 5. Spectral dependences of AOD: at the top shows situations with high aerosol turbidities of the atmosphere; and the bottom shows multiyear averages in two regions

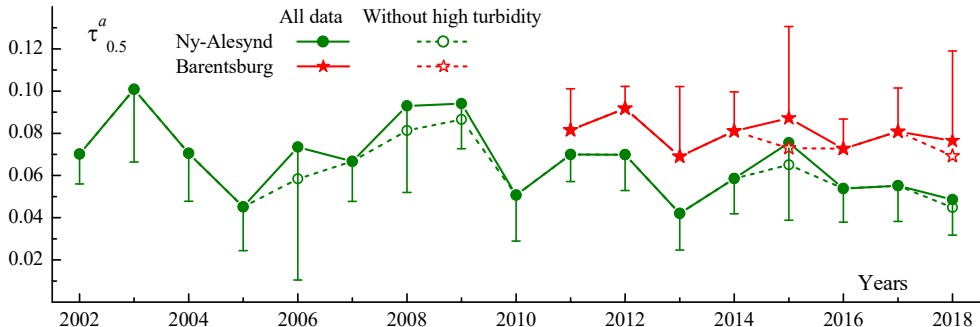

Fig. 6. Multiyear variations in Annual AOD in Ny-Ålesund and Barentsburg

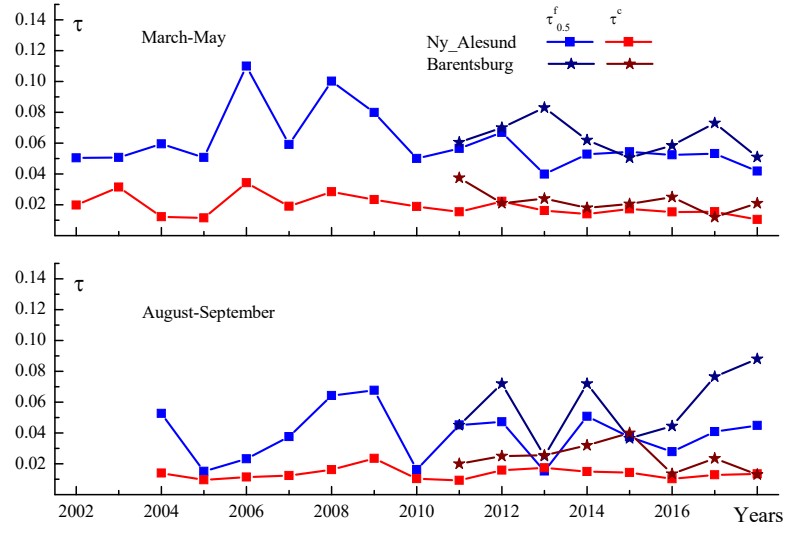


Fig. 7. Multiyear variations in $\tau_{0.5}^f$ and $\tau^c$ for the periods of spring maximum and fall minimum of AOD in two regions of Spitsbergen





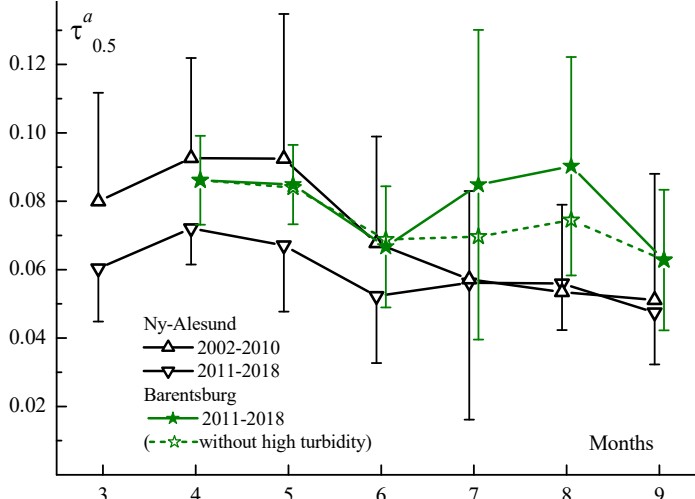

Fig. 8. Seasonally average $\tau^a_{0.5}$ variations in Ny-Ålesund and Barentsburg

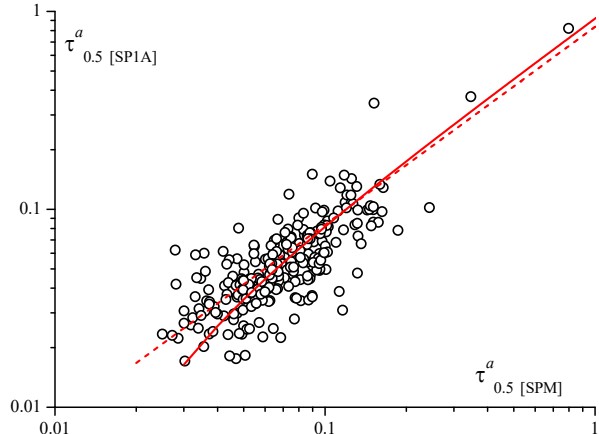

Fig. 9. Interrelation between Daily AOD (0.5 μm) values, measured in Ny-Ålesund and Barentsburg (2011-2018)

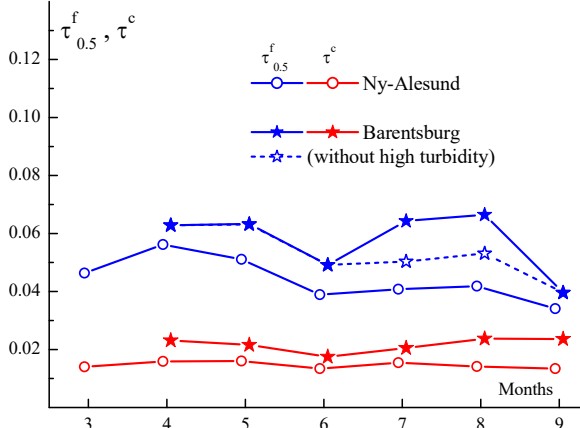

Fig. 10. Seasonal variations in fine and coarse AOD components in Ny-Ålesund and Barentsburg

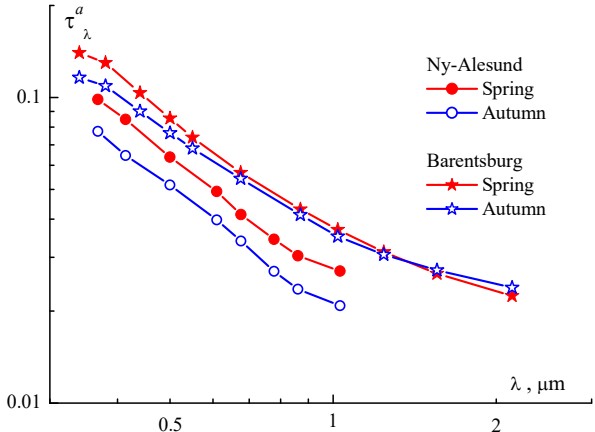

Fig. 11. Average spectral dependences of AOD in two regions of Spitsbergen (2011-2018) during spring (April-May) maximum and fall
(August-September) minimum

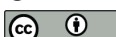



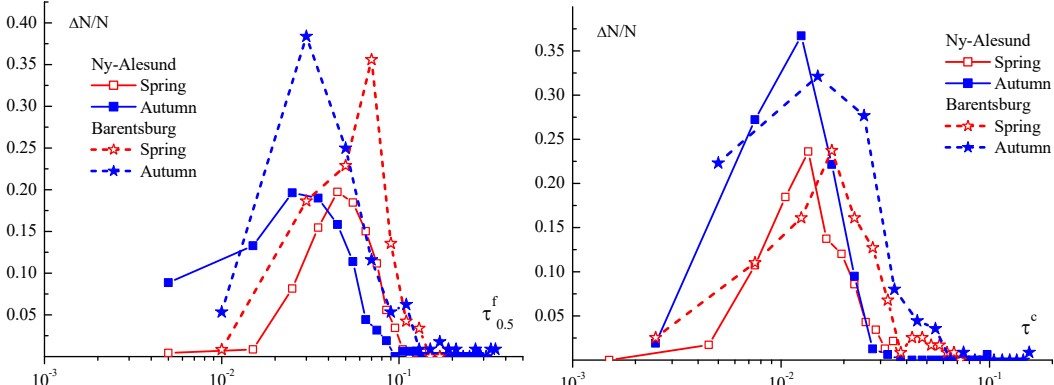

Fig. 12. Frequency histograms of (a) $\tau_{0.5}^{f}$ and (b) $\tau^{c}$ in spring and fall in two regions of Spitsbergen (2011-2018)

