# Peer review of "Interannual and seasonal variations in aerosol optical depth of the atmosphere in two regions of Spitsbergen Archipelago (2002-2018)"

_Atmospheric Measurement Techniques, 2020_

## Referee Comment (RC1) · Anonymous Referee #1 · 10 Jun 2020

General comments

The manuscript "Interannual and seasonal variations in aerosol optical depth of the atmosphere in two regions of Spitsbergen Archipelago (2002-2018) prepared by Dmitry M. Kabanov, Christoph Ritter, Sergey M. Sakerin raises very important issues connected to the climate change through the variations of its component, the measurements of Aerosol Optical Depth. This study is exceedingly relevant for climate variability in both global and regional scales, which in these times becomes extremely important. This manuscript still requires small and mostly editorial minor revision be-

fore publication. The article is concise and contains logical and thoughtful information. The paper is well organized – all chapters are written properly and point the attention of a reader towards an actual information. It also fits well according to the AMT scopes. The manuscript was written in proper English, also it does not contain too many language errors, and those that appear in the article are small and do not affect the substantive assessment of this work. I maintain that the manuscript will be clarified an rich enough for publication after including those corrections. The subject of Aerosol Optical Depth is quite often discussed in many articles due to the amount of reliable information that this parameter carries. Often, AOD information alone is an insufficient regional attribute, therefore related parameters, e.g. aerosol background or Angstrom parameter are added, or information not related to sun photometry (modeling results, information from satellite images or lidar data) are added. As I mentioned, this parameter is a very good characteristic, but photometric observations in the polar circle are very difficult and quite punctual, which is why I believe that the authors took up an important topic of long-term measurements - their seasonality and cyclicality. Over Spitsbergen, such measurements were already compared between Hornsund and Ny-Ålesund, as the authors mention at work, but the period of these comparisons concerned period. An important element of this work is to indicate that those comparison change a lot through years, it seasonal variations and episodic outflow determine the amplitude of interannual AOD variations in the Arctic atmosphere more by smokes from massive forest and agricultural fires. The authors also emphasize how important it is to analyze separately fine and coarse AOD components, having different spectral properties, origins, and lifetimes and how important is the character of the Arctic during AOD measurements, as demonstrated by the results from 2003, as imprecise values due to the insufficient number of measurements.

Specific comments about the substantive content:

Line 65 and 125: Citation needed

Line 135-140: After reading this paragraph, the question immediately arises, where

do such differences between stations derive from? This way of presenting and writing this paragraph causes that the reader is became less trustable to the authors. I propose that the authors reformulate the text, because this discrepancy between stations results both in measurements with two different instruments and in time shifting, i.e. the difference in time and direction of air mass inflow over a given station. For specific aerosol events, it is worth comparing the air backward trajectories to determine what is the time difference that the air mass data reached the various stations, which the authors often repeated in their other articles.

Line 165: The authors describe AOD very accurately, but they put the Angstrom formula into the text without first explaining. It would be worth harmonizing to make the article technically complete. Simple explanation will be satisfying.

Line 180: "and m and n are the parameters analogous to those in the Ångström formula." – please define the analogy, that is analogous to the $\beta$ and $\alpha$, which will make the text easier to follow for the readers.

185: In this paragraph it is worth highlighting what does it mean "independent data". This question arises immediately after the end of this paragraph, and the authors do not answer this question in a logical sequence, in the next paragraph.

Line 300: Is this predominance connected to the location of the station and orography, and also tendency that the direction of each outflow is less transformed at the north of Spitsbergen?

Fig 1 and 9: I would prefer adding a grid, even the same as on fig 3. It will facilitate reading it.

---

## Referee Comment (RC2) · Anonymous Referee #2 · 15 Jun 2020

The authors present a comparison of columnar optical aerosol characteristics measured at two arctic sites in Svalbard during a 16 years-long period. They report quasi-contemporaneous results from the two sites in terms of total, fine and coarse modes. The methods used to obtain the two modes are also discussed. The effect of forest-fire outbreaks are evaluated.

The paper can be considered of interest as a contribution to climatological studies on aerosol optical depth in this particular zone of the Arctic.

[Figure]

Specific comments

L14 "Tau fine" at the beginning of the sentence is a refuse?

L101 A conjunction is missing before the word "altitude".

L131 "Larger AOD values occur…" are these values the differences between different stations?

L143 Maximal -> maximum

L142-145 As the AOD usually decreases with increasing wavelengths, the same is obvious for the difference between different measurements. These differences should be normalized by the AOD value in the correspondent wavelength to estimate a relative contribution of fine and coarse particles.

L206 "by empirical method (EM)". It should be "an" or "the empirical method". Furthermore, in the rest of the text (e.g. L 387) you sometimes call it "base method" or "base empirical method". Please use always the same name, or better, the acronym EM.

L211 "using THE spectral deconvolution algorithm".

L215-216 Not clear to me: the averege difference ranges from 0.006 to 0.024? And the associated SD is 0.007?

L261 Please define "disperse composition"

L265-269 So you use IM1? You say you do not show the results for "aerosol dispersed composition" because they are more complex and dependent on the choices of the refractive index. But aren't the results of tauc and tauf0.5 equally dependent on it in this case (IM1)?

L303 Variations between the two periods?

Table3 Characteristics -> Parameters. Remove the empty line.

L335-337 So, in this comparison you are not using "near time coincident" values?

L359 How you detected the smoke?

Table4 Characteristics -> Parameters.

L373 The difference between 0.07 and 0.03 is 0.04, that is much higher of 0.015-0.016 that you report on the previous line. There is something I don't catch?

Figure 3. Caption: EA -> EM

Figure 9. What are the two red lines?

———————————————————————

---

## Author Comment (AC1) · 22 Jun 2020

we thank you for your constructive review and attach our answer to all raised questions.

Our modifications are:

Quote line 65: The warming of Spitsbergen and the possible relation to changed atmospheric circulation pattern was shown here: (Isaksen et al. 2016) https://agupubs.onlinelibrary.wiley.com/doi/full/10.1002/2016JD025606

[Figure]

The impact of the West Spitsbergen current on the local climate is described here https://rmets.onlinelibrary.wiley.com/doi/full/10.1002/joc.2338

We include this quote in the new version of the manuscript Isaksen, K.; Nordli, O.; Forland, E.J.; Lupikasza, E.; Eastwood, S. and Niedzwiedz, T. Recent warming on Spitsbergen—Influence of atmospheric circulation and sea ice cover, J. Geophys. Res. Atmos., 121, 11,913–11,931, doi: 10.1002/2016JD025606. 2016

Walczowski, W.; Piechura, J. Influence of the West Spitsbergen Current on the local climate. nt. J. Climatol. 31: 1088–1093, doi: 10.1002/joc.2338. (2011)

Quote line 125: The possibility to derive differences on aerosol properties comparing neighboring stations for Spitsbergen is described in the coming paragraph with corresponding quote. For this reason we include a more general quote here (Sakerin et al. 2010) https://link.springer.com/article/10.1134/S1024856010020028 The importance of Arctic intercomparison campaigns with error estimation was provided by Mazzola et al. 2011 https://repositorio.aemet.es/bitstream/20.500.11765/11692/1/Mazzola-Atm-Env-2011.pdf

We include this quote in the new version of the manuscript Sakerin S.M., Kabanov D.M., Nasrtdinov I.M., Turchinovich S.A., and Turchinovich Yu.S. The results of two-point experiments on the estimation of the urban anthropogenic effect on the characteristics of atmospheric transparency // Atmospheric and Oceanic Optics, 2010, Vol. 23, No. 2, p. 88–94. DOI: 10.1134/S1024856010020028.

Mazzola, M.; Stone, R.S.; Herber, A.; Tomasi, C.; Lupi, A.; Vitale, V.; Lanconelli, C., Toledano, C.; Cachorro, V.E.; O'Neill, N.T.; Shiobare, M.; Aaltonen, V. et al. Evaluation of sun photometer capabilities for retrievals of aerosol optical depth at high latitudes: The POLAR-AOD intercomparison campaigns. Atmosph. Environm. doi: 10.1016/j.atmosenv.2011.07.042. 2011

Paragraph lines 135 – 140: on the differences in AOD over both stations. In his works

[Toledano et al., 2012; Pakszys and Zielinski, 2017] had already compared AOD, measured at the neighboring stations Hornsund and Ny-Ålesund. The difference in the seasonally and annually average values of AOD between these stations reached 0.01-0.02, mainly due to situations with high atmospheric turbidities. We obtained about the same result after hourly average AOD in Ny-Ålesund and Barentsburg were compared (when measurements were within one hour of each other). How can the difference in AOD between the neighboring regions be explained? Even without local anthropogenic impact, the difference in AOD between stations, separated by mountains and distances 100 km (or longer) apart may be due to the Arctic Haze phenomena and transports of smoke plumes. (They are not conservative and homogeneous structures, blown by wind without changes). The Arctic Haze or plumes may be observed in the region of any station, and may be observed (to a lesser extent) or not (due to spatial inhomogeneities) at another one. When AOD, measured at two stations within an hour (as in our work), are compared, the number of such inhomogeneous situations can only be reduced partly sometimes, but can never be eliminated at all. Resort to data from trajectory analysis or to any correction (or shift) in time makes no sense for two reasons. First, this cannot be made due to spatial inhomogeneities in AOD: for instance, there was the thickest part of the plume in the region of one station, and a thinner part in the region of another one. Second, the AOD observations are not continuous, being carried out only in situations when Sun is not covered by clouds. That is, the measurements were in the period of AOD maximum at one station, and only during AOD decay or commencing AOD growth at another one. Taking into consideration the Reviewer's comment, we corrected slightly the text in this paragraph: "Comparison of measurements with the two photometers showed a large dispersion of the data under the conditions of strong atmospheric turbidities, namely, during outflow of smoke plumes from forest fires and in the Arctic Haze situations. Due to large spatial inhomogeneity of these structures, AOD, measured in two regions, may strongly differ, making the comparison incorrect".

Line 165: thanks, we added a short explanation to the Ångström formula in the new

manuscript: The attenuation of radiation by atmospheric aerosol varies as a function of wavelength, depending on sizes and refractive index of aerosol particles. To characterize the AOD, measured at different wavelengths, the Ångström formula is widely used: , (1) where $\beta$ and $\alpha$ are the approximation parameters of the spectral dependence of AOD; $\beta$ is the turbidity coefficient, which is close in value to AOD at the wavelength of 1 $\mu$m; and $\alpha$ is the selectivity exponent (power-law decay).

Line 180: Thanks – we explain now that m corresponds to $\beta$ and n to $\alpha$ in eq. 1. Line 185: Thanks we changed the wording: with independent data we mean additional information. We write: "But, precisely what caused changes in the selectivity of AOD is almost impossible to determine without the use of additional information like e.g. aerosol in-situ measurements".

Line 300: In this paragraph, we compare two periods of measurements at a single station (Ny-Ålesund) and indicate a tendency toward a small AOD decrease in 2011-2018 relative to 2002-2010. The location of this station, orography, or something else did not change. By the wording "no explicit predominance" we wanted to express that no single value for the variation coefficient for fine or coarse mode for the three places and times: Ny-Ålesund: early, Ny-Ålesund: later, Barentsburg: later dominate over the others. This means that there is no clear shift in aerosol properties neither in time nor from Barentsburg to Ny-Ålesund. We rewrite the sentence for clarity: No explicit predominance of the variation coefficients for any AOD component can be seen. The relative variations $\tau$f0.5 and  ÑĄ are about the same: their variation coefficients V are 14-29%. Neither AOD component shows a clear predominance of variation coefficients.

We fixed the Fig. 1 and 9, as suggested by the Reviewer.

Please also note the supplement to this comment: https://www.atmos-meas-tech-discuss.net/amt-2020-83/amt-2020-83-AC1-supplement.pdf

[Figure]

[Figure]

**Fig. 1.**

[Figure]

**Fig. 2.**

[Figure]

**Fig. 3.**

---

## Author Comment (AC2) · 22 Jun 2020

We thank the reviewer for the careful reading and the generally positive review. We answered all questions and remarks. See below and in the attachment:

L 14: we do not use a symbol in the abstract any longer

L 101: thanks – corrected

L 131 "Larger AOD values occur" Yes, this difference in AOD is between two stations,

not between seasonal or annual values, but between specific situations. For a more clarity, we corrected somewhat the sentence: "Large differences in AOD between these stations occur in periods of the Arctic haze and outflows of smokes from forest fires".

L 143: thanks – corrected!

L 142-145: decreasing AOD towards IR and variability This is a good remark. However, we considered this: In Fig 11 it will be shown later that the AOD at 0.38 $\mu$m is typically less than a factor of 4 larger than the AOD at 0.87$\mu$m, while the difference between the sites is almost a factor of 5. We add (new part in bold) This feature is real despite the decreasing AOD at longer wavelengths and indicates that fine aerosol is more abundant in the atmosphere of Barentsburg

L 206: Thanks we use always EM in the new version.

L 211: thanks – corrected

L215 – 216: We clarify that the average difference in ïĄť ÑĄ, calculated by different methods (and for different conditions), relative to the empirical method is, indeed, no higher than 0.007. While the standard deviations of the regression between the two compared ïĄť ÑĄ values are in the range of 0.006-0.024 (see [Kabanov et al., 2016] for more detail). L 261: disperse composition By disperse composition we mean the size distribution and changed the wording in the manuscript.

L 265 – 269: Here, there are two questions. 1. Yes, in this case we used IM1, which has much better characteristics of the interrelation with EM data. You are right: IM1 also depends on the choice of the refractive index. But this is not important because at the last stage of implementation of this method (see Line 239-240), we also use the regression relation and select the approximation parameters. That is, we could specify a slightly different refractive index and select slightly different approximation parameters for the linear regression. We made so and obtained about the same result. In principle, simpler methods of the $\tau$ÑĄ calculation could be used (e.g., RM2). From

Table 2 it can be seen that the errors of different methods differ insignificantly. 2. Of course, seasonal and interannual variations are easier to analyze in the optical characteristics: they are just two, $\tau\tilde{N}Ą$ and $\tau f$. Analysis of variations in microphysical composition of aerosol is a more complex problem: it will be necessary to consider the particle distribution functions (i.e., changes in two or three parameters for each aerosol fraction) and, moreover, the refractive index, which is at all unknown, in this case.

L 303: We corrected somewhat two sentences in this paragraph: "The relative variations in andïĄť ŇĄ are about the same: their variation coefficients V are 14-29%. Neither AOD component shows a clear predominance of variation coefficients".

Table 3: line deleted, thanks!

L 335 – 337: The seasonal and interannual AOD variations were analyzed individually over a full dataset in each region. That is, no selection of data with identical hours of measurements was performed in this case.

L 359: To identify smoke outflows (in the cases of large AOD values), we used data on back trajectories of air mass motion (HYSPLIT) in combination with satellite maps of fire centers (temperature anomalies).

Table 4: thanks - corrected

L 373: We clarify that the average and modal (most probable) values are different statistical characteristics. The first sentence of this paragraph is about the average values, presented in Table 4 and in Fig. 10. In turn, the second sentence is about modal values, which are presented in Fig. 12Đř. Thus, there is no contradiction in that the average values decrease by the amount 0.015-0.016, while modal values decrease from 0.07 to 0.03.

Figure Caption 3: thanks – corrected

Figure 9: The solid line is the unconstrained fit (Y=aX+b). The dotted line is the fit through the origin (Y*aX). We explain this in the new version.

Please also note the supplement to this comment:
https://www.atmos-meas-tech-discuss.net/amt-2020-83/amt-2020-83-AC2-
supplement.pdf

[Figure]

[Figure]

**Fig. 1.**

[Figure]

**Fig. 2.**

[Figure]

[Figure]

**Fig. 3.**

---

## Referee Comment (RC3) · Anonymous Referee #3 · 28 Jun 2020

General comments :

This article presents four methods (already published in previous works) to discriminate the fine and coarse mode components of the aerosol optical depth in a limited spectral range, for two locations, in Spitsbergen islands. One method (IM1) is then selected based on its performance with respect to a more accurate base method, using a larger spectral range. The subsequent inter and intra-annual analysis based on those aerosol modes follows the similar work published for the Tomsk region, pointing to the fact that summer time maximums is mainly due to sporadic smoke intrusions. While the article

looks more as an incremental step in the work of the authors, I think it represents a significant and necessary contribution in updating the aerosol dynamics knowledge in the Spitsbergen islands and in the Arctic in general. However, I think the sections describing the fine/coarse mode discrimination methods lack of clarity. It is difficult to understand and eventually reproduce them by a reader, particularly IM1 and IM2. They are relaying too much on information available on references hard or even impossible to find, like those on the Russian journal "Atmospheric and Oceanic Optics", unavailable online prior to 2009. Some details seem actually never published. I think this is the part where the paper needs the most improvement. My specific comments below are actually only refereeing to that part. While this may look at first as a potential major revision, since it can be easily fixed in a short time, I consider it minor revision.

Specific comments:

P01L27: The reason for 10 or 20 year periods is not enough explained. The best would be a reference. If not, may be based on the expected tendency and natural variability, one may figure out what would be a significant period.

P02L49: Stratospheric aerosol from Kasatochi (or later produced from volcanic SO2) lasted until the end of the year 2008, with some reminiscence at the beginning of the next year. I think pollution from every source going into stratosphere may last months, even smog from firestorms.

P05L142: "A comparison of the statistical characteristics showed that the average AOD values are a little larger in Barentsburg than in Ny-Ålesund". This cannot be concluded from Fig 1 without showing error bars on the linear fit.

P05L145: "At the same time, we note that the AOD differences are minor (comparable with the error)". As a reader, I have no idea what is the error at this stage.

P05L148: "$\Delta = 0.004 - 0.024$" is this the difference between Barentsburg and Hornsund? What is the error on such measurement? 0.004 is less than the possible achievable error (0.007 in Table 2). 0.004 is also 6 times less than 0.024 ! How can we trust such numbers?

P06L168: "Numerous studies… showed". Since it is a well-known concept, may be a textbook reference is appropriate.

P06L179: In the context of the equation (2) and related Fig 2, you may mention that the departure from linearity in Fig 2 may be due to a second order behavior of alpha, as used in the SDA, but ignored in this simpler approach. You may however suggest that this simpler method may be less prone to error propagation.

P07L191: "P < 0.0001" – P was never defined. Even if for some people it may be a well known parameter, it's not necessarily obvious what it is. You need to briefly mention what it is and may give a reference for further details, like a textbook.

P07L199: "The lifetime of fine aerosol is a few days". As mentioned previously, stratospheric aerosol can last longer than those in the troposphere. So, one should precise here what aerosols are we talking about.

P08L214: "$\tau^f\_0.5 (\tau_c)$ " what does it mean, $\tau^f\_0.5$ as a function of $\tau_c$ ? It's not really obvious! Please explain!

P08L217: Reference [Kabanov et al., 2019], is it "a" or "b" ?

P08L221: It took me a long time to understand what RM1 and RM2 are! Most readers do not have access to your [Kabanov et al., 2019] and particularly [Kabanov and Sakerin, 2016] references, without which is impossible to understand. I think you need to provide a minimal info to facilitate the understanding of what are you are doing! Actually, in your text is not even mentioned that you need specifically to check [Kabanov and Sakerin, 2016] in order to understand RM1 and RM2. Also, you should mention that Fig 3 is a reproduction of Fig 2 from [Kabanov et al., 2019].

P08L228: Table 2 should be located after you introduce IM1 and IM2! Why did you changed from "$\tau^f\_0.5 (\tau_c)$" to "$\tau_c(\tau^f\_0.5)$" ? Why do you have a space

below \tauˆf_0.5?

P08L230: You need to mention [Kabanov and Sakerin, 2016] section 3, in order to justify "conditions of a specific region".

P09L238: You need to provide much more info about the IM1 and IM2 methods, to a point that a reader can reproduce your work. For example, dV/dr is not even mentioned in your only reference [Sviridenkov, 2001] with respect to those calculations. Actually, you should not even assume that the reader can consult [Sviridenkov, 2001], as the "Atmospheric and Oceanic Optics" references are not available online prior to 2009 (and older hard-copies are probably available only in Russia).

P09L256: 0,086 should be 0.086.

---

## Author Comment (AC3) · 7 Jul 2020

We thank the reviewer for the careful review and helpful remarks. As you can see, we answered all questions and comments. Our answers are:

**Line 27**.

*In the phrase about 10(20) years there is no need in making a reference: it is just a reminder about the well-known fundamentals of statistical data analysis. To identify the regularities of variations in climate characteristics, which depend upon many processes on various variability scales, the length of observation time series should be many times (an order of magnitude or more) larger than the periodicity sought. For instance, to determine the character of the seasonal behavior, the length of observation time series should be of the order of 10 years. This will be shown later in our section 3.1, figure 6. However, under the conditions of reforming climate system, the statistical estimates may give a distorted or incomplete understanding of the studied process because of the new variability factors emerged. Taking into consideration your comment, we modified one sentence (Line 28-29) a little:*
"However, under the conditions of changing climate system, a time series 10 years (or even 20 years) long may turn out to be insufficient to identify correctly the tendencies or periodicities in variations of aerosol characteristics".

**Line 49**.

*You are probably right. The impact of the Kasatochi eruption on sun-photometer data only appeared short as the season of observations ended early Oct. We rewrote the sentence:*

new

The effects of less powerful volcanic eruptions on the Arctic atmosphere are short-term to mid-term (some weeks in duration). For instance, AOD increase on Spitsbergen was observed after the eruptions of volcanoes Kasatochi (August / Sep 2008) and Sarychev (after July 2009) [Hoffmann et al., 2010; Toledano et al., 2012].

old

The effects of less powerful volcanic eruptions on the Arctic atmosphere are short-term (a few days in duration) and comparable to those due to smokes from forest fires. For instance, AOD increase on Spitsbergen was observed after the eruptions of volcanoes Kasatochi (August 2008) and Sarychev (July 2009) [Hoffmann et al., 2010; Toledano et al., 2012].

**Line 142**.
(a) *This phrase does not refer directly to Fig. 1, because it is in the next paragraph. Difference in the data from quasi-synchronous AOD measurements between two stations was already considered in our previous work [Kabanov et al., 2018] (see reference citation in Line 134), so only a conclusion is presented here. For more clarity, we gave the reference citation again in this sentence:*
"A comparison of the statistical characteristics showed that the average AOD values are a little larger in Barentsburg than in Ny-Ålesund [Kabanov et al., 2018]".

(b) *We think that it is a bad idea to display the average difference in the data between two stations in Fig. 1. But, if the Referee considers that this is mandatory, we prepared the following variant of the figure:*

[Figure]

Fig. 1. Scatter diagram of AOD measurements in two regions using SP1A (Ny-Ålesund) and SPM (Barentsburg) photometers. Solid line: unconstrained linear regression; dotted line: regression through origin (0,0); green indicates the average value of the data difference and the standard deviation

**Line 145**. *Taking into consideration your comment, we modified one sentence (the word **error** was replaced by **uncertainty** and links were added):*

«At the same time, we note that the AOD differences are minor (comparable with uncertainty of determining AOD – about 0.01-0.02 [Kabanov et al. 2009; Sakerin et al., 2013]), and the interdiurnal AOD variations in the two regions are coordinated in character (correlation coefficients are 0.83-0.89)».

**Line 148**.

(a) *Yes - the $\Delta$ symbol indicates the average difference between the measurement data of two photometers (or AOD at two stations) at different wavelengths. This is not a measurement error, but the difference in physical characteristics in the two regions. The error (or rather, uncertainty) of the AOD measurements is 0.01-0.02 (see Line 145).*

(b) *Table 2 shows statistical characteristics on a completely different issue: standard deviations $\sigma$ and correlation coefficients R between $\tau^c$ values calculated by different methods.*

**Line 168**. *Although Angstrom's formula is well known, we have added links at the request of the Referee:*

«Numerous studies in different regions and atmospheric conditions showed that the formula (1) does describe well the wavelength dependence $\tau^a(\lambda)$ in the main range (0.34 – 1 μm) of AOD measurements [Angstrom, 1964; Shifrin, 1995; Eck et al., 1999; Cachorro et al., 2000; Schuster et al., 2006].

1. Angstrom A. Parameters of atmospheric turbidity // Tellus XVI. 1964, N 1, p. 64-75.
2. Shifrin K.S. Simple relationships for the Angstrom parameter of disperse systems // Appl. Opt., 1995, Vol. 34, 4480-4485.
3. Eck, T.F., Holben B.N., Reid J.S., Dubovik O., Smirnov A., O'Neill N.T., Slutsker I., and Kinne S. Wavelength dependence of the optical biomass burning, urban, and desert dust aerosol // J. Geophys. Res., 1999, Vol. 104, 31333-31350.
4. Cachorro V.E., Duran P., Vergaz R. and de Frutos A.M. Measurements of the atmospheric turbidity of the north-centre continental area in Spain: spectral aerosol optical depth and Angstrom turbidity parameters // J. Aerosol Sci. 2000, Vol. 31, No. 6, pp. 687-702.
5. Schuster, G.L., Dubovik O., and Holben B.N. Angstrom exponent and bimodal aerosol size distributions // J. Geophys. Res., 2006, Vol. 111, D07207. doi:10.1029/2005JD006328.

**Line 179**.

*The main point in this section was to show that the Ångström parameters are inconvenient for analysis of the causes for AOD variations. In particular, the parameter $\alpha$ depends on the relationship between two AOD components ($\tau^f$ and $\tau^c$). Therefore, based on the parameter $\alpha$, it is difficult to judge what (whether $\tau^f$ or $\tau^c$) was the cause for the AOD variations.*

*The dependence of $\alpha$ on the components $\tau^f$ and $\tau^c$ is more complex and explicitly nonlinear in character. A variant of this dependence in the form $\alpha = ln[(\tau_{0.5}^f/\tau^c) + 1]$ is presented in Fig. 2. Seemingly, a better approximation of this dependence can be selected; for instance, "a second order behavior of alpha" can*

*be used, as was done in the SDA method. However, this requires a separate study, and was beyond the scope of this paper. In this case, it was sufficient to show that $\alpha$ does depend on the relationship ($\tau^f/\tau^c$), and we did so (see regression in Fig. 2).*

**Line 191**. *An explanation of the symbol P (confidence level) was added to this sentence:*
«The correlation coefficient between $\alpha$ and $\ln[(\tau_\lambda^f/\tau^c)+1]$ is statistically significant and equal to 0.68 (confidence level P < 0.0001)».
*"Confidence level P" is a standard and well-known statistical characteristic. Therefore, there is no need in providing a reference to any text-book.*

**Line 199**. *In this sentence, we have added refinement (in troposphere):*
«The lifetime of fine aerosol in troposphere is a few days; therefore, it can be transported long distances (hundreds and thousands of kilometers) away».

**Line 214**. *You are right: the notation $\tau_{0.5}^f$ ($\tau^f$) may be unclear. Therefore, in this sentence and in the text below we made the following correction: $\tau_{0.5}^f$ (or $\tau^f$). We clarify that, within different methods, the researchers calculate anyway a single component (either $\tau_{0.5}^f$ or $\tau^c$), while the other is found as a residual of the total AOD (see Line 207-208).*

**Line 217**. *Thank you for your comment. We fixed the link (correct – [Kabanov et al., 2019**a**]):*
«For the conditions of Arctic region (Spitsbergen), we performed an additional study [Kabanov et al., 2019a], concerning the selection of an optimal method of $\tau_{0.5}^f$ (or $\tau^c$) estimation».

**Line 221**. *According to the Referee's comment, we added two clarifying sentences (see paragraph above):*
" … In the first regression method (RM1), $\tau^c$ is estimated using its interrelation with the parameter $\beta$ (see formula (3) below). In the second method (RM2), the regression dependence of $\tau_{0.5}^f$ on the parameters $\alpha$ and $\beta$ (see formula (4)) is used. Comparison of different methods …"

**Line 228**: *Thanks, we shifted the Table 2 and corrected the wrong layout.*

**Line 230**. *We added the link [Kabanov and Sakerin, 2016], as suggested by the Referee:*
«The disadvantage of the regression methods is that they require a preliminary data accumulation under the conditions of a specific region for determining optimal regression coefficients in equations (3) and (4) [Kabanov and Sakerin, 2016]».

**Line 238**.
*It is not quite clear which part of the IM1 method is requested by the Referee to be described in a more detail. We will try to do so, but before we will give three clarifications.*
*1. In implementing IM1 (or IM2) it is entirely immaterial which (Sviridenkov's or another) method of solving the inverse problems is to be used. A large number of methods have been developed in the last half-century to retrieve the particle distribution functions from spectral measurements of AOD or aerosol extinction coefficients. That is, any accessible and verified algorithm of solving the inverse problem, making it possible to determine the distribution function (dV/dr), or (dS/dr), will be appropriate.*
*2. It is immaterial what (whether dV/dr or dS/dr) should be retrieved, because an interrelation with the optical characteristic, $\tau^f$ or $\tau^c$, can be found for any of them. In addition, (dV/dr) and (dS/dr) are interrelated mathematically (either can be calculated if the other is known). Therefore, only one characteristic, (dV/dr), will be used below.*
*3. In this work, we used the (available to us) Sviridenkov's algorithm of solving the inverse problem, which is a modification of the Twitty algorithm. The algorithm of the (now died) Sviridenkov M.A. has been tested and successfully used for as long as 20 years by a few scientific groups from Russia.*

*Because you encountered problems with the reference to the work of Sviridenkov, we asked our Editorial office to make this issue (No. 12, 2001) of the Journal publicly available. We note that the results, obtained using the Sviridenkov's algorithm (as a part of the IM1 and IM2 methods), well agree with the data from other techniques (EM, RM1, RM2, and SDA). Therefore, we have no grounds to not trust the Sviridenkov's algorithm.*

*Using the IM1 method as an example, we will explain the consecutive steps of its implementation.*

*1st step. Based on any known method of solving the inverse problem (for a specified refractive index, type of the particle distribution function, and grid of radius ranges), the spectral AOD values are used to calculate the particle distribution function (dV/dr) or (dS/dr).*

*2nd step. In the distribution (dV/dr) thus obtained we select its part referring to the fine (submicron) fraction, and, for it, calculate the total particle volume ($V^f$) through integration. The size (radius) integration limits for particles in the fine fraction are: left boundary is specified depending on specific features of the chosen inversion algorithm (usually $r \approx 0.1$ μm); and 0.4 or 0.5 μm is taken as the right boundary.*

*3rd step. We consider the regression interrelation between particle volumes in the fine fraction ($V^f$) and the $\tau^f$ values, calculated by the empirical method (EM). The interrelation thus obtained (Fig. 4a) is used to select the parameters of a linear regression equation which makes it possible to calculate the component $\tau^f$ according to the particle volumes $V^f$. (Figure 4b illustrates the comparison of $\tau^f$ values, obtained by the two (EM and IM1) methods).*

*Taking into consideration the Referee's comments, we provided the text about the IM1 method in a little more detail (two insertions):*

*(1) "This method is implemented in the following steps.*

*1st step. Based on any known method of solving the inverse problem (for a specified refractive index, type of the particle distribution function, and grid of radius ranges), the spectral AOD values are used to calculate the particle distribution function (dV/dr) or (dS/dr).*

*2nd step. In the distribution (dV/dr) thus obtained we select its part referring to the fine fraction, and, for it, calculate the total particle volume ($V^f$) through integration.*

*3rd step. We consider the regression interrelation between particle volumes in the fine fraction ($V^f$) and the $\tau^f$ values, calculated by empirical method (EM). The interrelation thus obtained (Fig. 4a) is used to select the parameters of a linear regression equation which makes it possible to calculate the component $\tau^f$ according to the particle volumes $V^f$".*

*(2) "The inverse problem on retrieving the distribution functions (dS/dr) was solved using iteration algorithm of M.A. Sviridenkov [Sviridenkov, 2001], modified from Twitty algorithm [Twitty, 1975]. The particle distribution was assumed to be lognormal, and the refractive index was assumed to have the real part of 1.5 and the imaginary part of 0. In the calculations we used the following radius grid: 0.09-0.13-0.17-0.21-0.25-0.29-0.33-0.37-0.41-0.45-0.49-0.53-0.59-0.65-0.81-0.99-1.21-1.59-1.81-2-2.5-3 μm".*

**Line 256**. *Thank you, corrected!*

---

## Author Comment (AC6) · 22 Jul 2020

We thank the reviewer for the careful review and helpful remarks. As you can see, we answered all questions and comments. Our answers are:

Line 27. In the phrase about 10(20) years there is no need in making a reference: it is just a reminder about the well-known fundamentals of statistical data analysis. To identify the regularities of variations in climate characteristics, which depend upon many processes on various variability scales, the length of observation time series should be

many times (an order of magnitude or more) larger than the periodicity sought. For instance, to determine the character of the seasonal behavior, the length of observation time series should be of the order of 10 years. This will be shown later in our section 3.1, figure 6. However, under the conditions of reforming climate system, the statistical estimates may give a distorted or incomplete understanding of the studied process because of the new variability factors emerged. Taking into consideration your comment, we modified one sentence (Line 28-29) a little: "However, under the conditions of changing climate system, a time series 10 years (or even 20 years) long may turn out to be insufficient to identify correctly the tendencies or periodicities in variations of aerosol characteristics".

Line 49. You are probably right. The impact of the Kasatochi eruption on sunphotometer data only appeared short as the season of observations ended early Oct. We rewrote the sentence: new The effects of less powerful volcanic eruptions on the Arctic atmosphere are short-term to mid-term (some weeks in duration). For instance, AOD increase on Spitsbergen was observed after the eruptions of volcanoes Kasatochi (August / Sep 2008) and Sarychev (after July 2009) [Hoffmann et al., 2010; Toledano et al., 2012]. old The effects of less powerful volcanic eruptions on the Arctic atmosphere are short-term (a few days in duration) and comparable to those due to smokes from forest fires. For instance, AOD increase on Spitsbergen was observed after the eruptions of volcanoes Kasatochi (August 2008) and Sarychev (July 2009) [Hoffmann et al., 2010; Toledano et al., 2012].

Line 142. (Ðř) This phrase does not refer directly to Fig. 1, because it is in the next paragraph. Difference in the data from quasi-synchronous AOD measurements between two stations was already considered in our previous work [Kabanov et al., 2018] (see reference citation in Line 134), so only a conclusion is presented here. For more clarity, we gave the reference citation again in this sentence: "A comparison of the statistical characteristics showed that the average AOD values are a little larger in Barentsburg than in Ny-Ålesund [Kabanov et al., 2018]". (b) We think that it is a bad idea

to display the average difference in the data between two stations in Fig. 1. But, if the Referee considers that this is mandatory, we prepared the following variant of the figure:

Fig. 1. Scatter diagram of AOD measurements in two regions using SP1A (Ny-Ålesund) and SPM (Barentsburg) photometers. Solid line: unconstrained linear regression; dotted line: regression through origin (0,0); green indicates the average value of the data difference and the standard deviation

Line 145. Taking into consideration your comment, we modified one sentence (the word error was replaced by uncertainty and links were added): ÂńAt the same time, we note that the AOD differences are minor (comparable with uncertainty of determining AOD – about 0.01-0.02 [Kabanov et al. 2009; Sakerin et al., 2013]), and the interdiurnal AOD variations in the two regions are coordinated in character (correlation coefficients are 0.83-0.89)Âż.

Line 148. (Đř) Yes - the $\Delta$ symbol indicates the average difference between the measurement data of two photometers (or AOD at two stations) at different wavelengths. This is not a measurement error, but the difference in physical characteristics in the two regions. The error (or rather, uncertainty) of the AOD measurements is 0.01-0.02 (see Line 145). (b) Table 2 shows statistical characteristics on a completely different issue: standard deviations  and correlation coefficients R between  ŃĄ values calculated by different methods.

Line 168. Although Angstrom's formula is well known, we have added links at the request of the Referee: ÂńNumerous studies in different regions and atmospheric conditions showed that the formula (1) does describe well the wavelength dependence  Đř() in the main range (0.34 – 1 ïĄ■m) of AOD measurements [Angstrom, 1964; Shifrin, 1995; Eck et al., 1999; Cachorro et al., 2000; Schuster et al., 2006]. 1. Angstrom A. Parameters of atmospheric turbidity // Tellus XVI. 1964, N 1, p. 64-75. 2. Shifrin K.S. Simple relationships for the Angstrom parameter of disperse systems //

Appl. Opt., 1995, Vol. 34, 4480-4485. 3. Eck, T.F., Holben B.N., Reid J.S., Dubovik O., Smirnov A., O'Neill N.T., Slutsker I., and Kinne S. Wavelength dependence of the optical biomass burning, urban, and desert dust aerosol // J. Geophys. Res., 1999, Vol. 104, 31333-31350. 4. Cachorro V.E., Duran P., Vergaz R. and de Frutos A.M. Measurements of the atmospheric turbidity of the north-centre continental area in Spain: spectral aerosol optical depth and Angstrom turbidity parameters // J. Aerosol Sci. 2000, Vol. 31, No. 6, pp. 687-702. 5. Schuster, G.L., Dubovik O., and Holben B.N. Angstrom exponent and bimodal aerosol size distributions // J. Geophys. Res., 2006, Vol. 111, D07207. doi:10.1029/2005JD006328.

Line 179. The main point in this section was to show that the Ångström parameters are inconvenient for analysis of the causes for AOD variations. In particular, the parameter  depends on the relationship between two AOD components (f and ÑĄ). Therefore, based on the parameter , it is difficult to judge what (whether f or ÑĄ) was the cause for the AOD variations. The dependence of  on the components f and ÑĄ is more complex and explicitly nonlinear in character. A variant of this dependence in the form  = ln[( / ÑĄ) + 1] is presented in Fig. 2. Seemingly, a better approximation of this dependence can be selected; for instance, "a second order behavior of alpha" can be used, as was done in the SDA method. However, this requires a separate study, and was beyond the scope of this paper. In this case, it was sufficient to show that  does depend on the relationship (f/ÑĄ), and we did so (see regression in Fig. 2).

Line 191. An explanation of the symbol P (confidence level) was added to this sentence: ÂńThe correlation coefficient between  and ln[( / ) + 1] is statistically significant and equal to 0.68 (confidence level P < 0.0001)Âż. "Confidence level P" is a standard and well-known statistical characteristic. Therefore, there is no need in providing a reference to any text-book.

Line 199. In this sentence, we have added refinement (in troposphere): ÂńThe lifetime of fine aerosol in troposphere is a few days; therefore, it can be transported long

distances (hundreds and thousands of kilometers) awayÂż.

Line 214. You are right: the notation (ŃĄ) may be unclear. Therefore, in this sentence and in the text below we made the following correction: (or ŃĄ). We clarify that, within different methods, the researchers calculate anyway a single component (either or ŃĄ), while the other is found as a residual of the total AOD (see Line 207-208).

Line 217. Thank you for your comment. We fixed the link (correct – [Kabanov et al., 2019Đř]): ÂńFor the conditions of Arctic region (Spitsbergen), we performed an additional study [Kabanov et al., 2019Đř], concerning the selection of an optimal method of (or ŃĄ) estimationÂż.

Line 221. According to the Referee's comment, we added two clarifying sentences (see paragraph above): " . . . In the first regression method (RM1),  ŃĄ is estimated using its interrelation with the parameter $\beta$ (see formula (3) below). In the second method (RM2), the regression dependence of on the parameters $\alpha$ and $\beta$ (see formula (4)) is used. Comparison of different methods . . ."

Line 228: Thanks, we shifted the Table 2 and corrected the wrong layout.

Line 230. We added the link [Kabanov and Sakerin, 2016], as suggested by the Referee: ÂńThe disadvantage of the regression methods is that they require a preliminary data accumulation under the conditions of a specific region for determining optimal regression coefficients in equations (3) and (4) [Kabanov and Sakerin, 2016]Âż.

Line 238. It is not quite clear which part of the IM1 method is requested by the Referee to be described in a more detail. We will try to do so, but before we will give three clarifications. 1. In implementing IM1 (or IM2) it is entirely immaterial which (Sviridenkov's or another) method of solving the inverse problems is to be used. A large number of methods have been developed in the last half-century to retrieve the particle distribution functions from spectral measurements of AOD or aerosol extinction coefficients.

That is, any accessible and verified algorithm of solving the inverse problem, making it possible to determine the distribution function (dV/dr), or (dS/dr), will be appropriate. 2. It is immaterial what (whether dV/dr or dS/dr) should be retrieved, because an interrelation with the optical characteristic, f or ÑĄ, can be found for any of them. In addition, (dV/dr) and (dS/dr) are interrelated mathematically (either can be calculated if the other is known). Therefore, only one characteristic, (dV/dr), will be used below. 3. In this work, we used the (available to us) Sviridenkov's algorithm of solving the inverse problem, which is a modification of the Twitty algorithm. The algorithm of the (now died) Sviridenkov M.A. has been tested and successfully used for as long as 20 years by a few scientific groups from Russia. Because you encountered problems with the reference to the work of Sviridenkov, we asked our Editorial office to make this issue (No. 12, 2001) of the Journal publicly available. We note that the results, obtained using the Sviridenkov's algorithm (as a part of the IM1 and IM2 methods), well agree with the data from other techniques (EM, RM1, RM2, and SDA). Therefore, we have no grounds to not trust the Sviridenkov's algorithm. Using the IM1 method as an example, we will explain the consecutive steps of its implementation. 1st step. Based on any known method of solving the inverse problem (for a specified refractive index, type of the particle distribution function, and grid of radius ranges), the spectral AOD values are used to calculate the particle distribution function (dV/dr) or (dS/dr). 2nd step. In the distribution (dV/dr) thus obtained we select its part referring to the fine (submicron) fraction, and, for it, calculate the total particle volume (Vf) through integration. The size (radius) integration limits for particles in the fine fraction are: left boundary is specified depending on specific features of the chosen inversion algorithm (usually r $\approx$ 0.1 $\mu$m); and 0.4 or 0.5 $\mu$m is taken as the right boundary. 3rd step. We consider the regression interrelation between particle volumes in the fine fraction (Vf) and the f values, calculated by the empirical method (EM). The interrelation thus obtained (Fig. 4Đř) is used to select the parameters of a linear regression equation which makes it possible to calculate the component f according to the particle volumes Vf. (Figure 4b illustrates the comparison of f values, obtained by the two (EM and IM1) methods).

Taking into consideration the Referee's comments, we provided the text about the IM1 method in a little more detail (two insertions): (1) "This method is implemented in the following steps. 1st step. Based on any known method of solving the inverse problem (for a specified refractive index, type of the particle distribution function, and grid of radius ranges), the spectral AOD values are used to calculate the particle distribution function (dV/dr) or (dS/dr). 2nd step. In the distribution (dV/dr) thus obtained we select its part referring to the fine fraction, and, for it, calculate the total particle volume (Vf) through integration. 3rd step. We consider the regression interrelation between particle volumes in the fine fraction (Vf) and the f values, calculated by empirical method (EM). The interrelation thus obtained (Fig. 4Đř) is used to select the parameters of a linear regression equation which makes it possible to calculate the component f according to the particle volumes Vf". (2) "The inverse problem on retrieving the distribution functions (dS/dr) was solved using iteration algorithm of M.A. Sviridenkov [Sviridenkov, 2001], modified from Twitty algorithm [Twitty, 1975]. The particle distribution was assumed to be lognormal, and the refractive index was assumed to have the real part of 1.5 and the imaginary part of 0. In the calculations we used the following radius grid: 0.09-0.13-0.17-0.21-0.25-0.29-0.33-0.37-0.41-0.45-0.49-0.53-0.59-0.65-0.81-0.99-1.21-1.59-1.81-2-2.5-3 $\mu$m".

Line 256. Thank you, corrected!
* * *
[Figure]

[Figure]

**Fig. 1.**

[Figure]

**Fig. 2.**

[Figure]

[Figure]

**Fig. 3.**

---

## Author Response (AR2)

Dear Andrew,

we thank you for your careful reading, corrections and remarks. Together with Dmitry Kabanov and Sergey Sakerin we have corrected the manuscript. The decline in fine mode AOD (at 500nm) is 0.016 within 10 years (with a confidence level of 0.062). We included these numbers in the new manuscript.

Thank you very much for your support.

Best regards

Christoph

Clean, final version:

[revised manuscript text omitted]

**logo of Copernicus Publications.**